# A Study on the Influence of Green Industrial Policy on Urban Green Development: Based on the Empirical Data of Ecological Industrial Park Pilot Construction

Xiaoyu He and Bo Li *

School of Finance and Public Administration, Anhui University of Finance and Economics, Bengbu 233030, China; hxy@aufe.edu.cn
* Correspondence: 3202000005@aufe.edu.cn; Tel.: +86-15755204851

**Abstract:** Balancing economic development and ecological protection is a dilemma that requires a solution. The construction of ecological industrial parks is expected to be the key to resolving this situation. Based on panel data from 276 prefecture-level cities in China spanning from 2004 to 2019, this paper presents a multi-period difference-in-differences (DID) model to identify the causal relationship between eco-industrial parks and the development of urban green spaces. The research indicates that the creation of eco-industrial parks can significantly promote the sustainable development of urban areas, with a policy promotion effect of approximately 0.0279. The analysis of the mechanism indicates that the implementation of a green industrial policy can enhance the level of sustainable development in cities. This can be accomplished by promoting eco-friendly innovation, facilitating the optimization of industrial structures, and strengthening environmental regulations. A heterogeneity analysis reveals that the impact of eco-industrial parks on promoting green development is more significant in cities located in the eastern and northern regions, as well as those with abundant human capital and financial resources. Conversely, cities situated in the central and western regions, as well as those with low levels of financial resources and non-human capital, tend to experience limited benefits from policies. The conclusions of this study can provide guidance for urban transformation and facilitate sustainable development. Moreover, these parks can function as case studies and provide valuable reference experiences for countries that have not yet established eco-industrial parks.

**Keywords:** eco-industrial park; green development; DID; sustainability; quality of life





## 1. Introduction

Since the reform and opening-up policy, China has maintained a relatively high economic growth rate over a long period, which is known as "China's growth miracle" [1]. It is widely accepted that China's local governments have entered into fierce competition for economic development as a response to the dual stimuli of China's decentralization of authority (political centralization of power and financial decentralization of power) and the "championship" of the political promotion of local officials [2]. Although such spontaneous competition among local governments has facilitated the rapid development of the local economy, it mainly depends on high consumption and a high input of energy and forms the "extensive development mode", which uses element driving and investment driving as the main growth engines [3,4]. To attract capital inflows and develop the economy under the extensive development mode, local governments often choose relatively relaxed environmental supervision, with a serious influence on the local ecological environment [3]. In a manner of speaking, the intensifying ecological environmental pressure is becoming an important influencing factor that restricts China's ability to improve people's welfare and quality of life and facilitate high-quality economic development [5]. As ecological civilization and green development are put on the agenda, regional development hopes to

gradually remove the traditional, single, economic-oriented mode and emphasize green, coordination and sustainability [6,7]. It has been highlighted in the report of the 20th CPC National Congress that "nature is the basic condition for human survival and development, and respecting, adapting to, and protecting nature is an inherent requirement for comprehensively building a modern socialist country". Promoting green development and facilitating the harmonious development of humans and nature is an important topic at present, which urgently needs to be solved.

Since entering the 21st century, as a new organizational model of the green development concept in the industrial field, eco-industrial parks have become an important way to realize ecological environmental protection and sustainable economic development _EN-REF_4 [8–10]. Compared with traditional parks, eco-industrial parks emphasize the green, low-carbon, and circular development mode, which helps to speed up the transformation of industrial parks to an ecological civilization and promotes the rapid transformation of industries from the old growth mode to the sustainable development mode. At the same time, eco-industrial parks can organically combine economic growth and environmental protection to improve the quality of economic growth and drive green development [11]. In addition, eco-industrial parks have established mandatory regulations on the pollution emissions of enterprises in the park. According to Porter's hypothesis, this helps to force enterprises to carry out green innovation, which is regarded as an effective policy and channel to achieve net-zero emissions by 2050 [12], and is conducive to maintaining the balance between the benefits of economic growth and the environmental costs brought by natural resources [13] and achieve the dual goals of development and environmental protection [14]. The concept of the eco-industrial park first developed from the "industrial symbiosis" of Kalenburg, Denmark [15], and the essence of the idea is that enterprises can establish a cooperative relationship regarding the use of byproducts. Subsequently, eco-industrial parks began to deliberately imitate natural ecological systems in their development to increase the utilization of resources and energy sources, decrease pollutant emissions, and improve the ecological environment [16]. Developed countries, including Denmark, America, UK, Japan and Australia, have all set up eco-industrial parks, which have promoted local ecological environmental protection and economic development [17,18]. At present, eco-industrial parks have become a reconstruction and construction direction of industrial parks in developed countries. Compared to foreign countries, China started later in terms of the development of eco-industrial parks, but China also established some trials [19]. For example, the Ministry of Ecological Environment, Ministry of Commerce, and Ministry of Science and Technology published measures and regulations concerning the construction and management of eco-industrial parks in 2003, 2007, and 2015, respectively. They proposed explicit requirements regarding the application and construction, acceptance inspection, naming, supervision, and management of eco-industrial parks. In 2015, the Ministry of Ecological Environment issued the official National Standard for Eco-industrial Demonstration Parks, which covers 32 indexes of economic development, industrial intergrowth, resource-saving, environmental protection and information disclosure. This standard explicitly regulates the number of projects in the new eco-industrial chain, the proportion of energy use, and the utilization of industrial wastewater in eco-industrial parks. Moreover, it proposes strict control of key pollutant emissions in the park and requires key enterprises to adopt clean production. It can be seen that ecological industrial parks while considering economic development, emphasize the protection of the ecological environment. Their reasonable construction and application will help China to reduce the emission of pollutants such as greenhouse gases and accelerate the realization of China's goal of building a resource-saving and environmentally friendly society [20]. China is the best-developing economy among the BRICS countries; the formulation and implementation of these green industrial policies will not only help BRICS countries to learn from and learn from each other, so as to lead BRICS countries towards sustainable development, but, more importantly, as BRICS countries currently contribute about 37% of the global GDP, this proportion is expected to reach 50% by 2030 [12,21]. BRICS countries are also listed

as one of the largest greenhouse-gas-emitters in the world, so the transformation of their industrial policies is more conducive to global sustainable development [14].

Based on the basic fact that eco-industrial parks have the attribute of environmental protection, this study focuses on influences that the green industrial policies of eco-industrial parks have on the green development of cities. Using systematic and strict empirical studies, this study tried to answer the following core questions, which have no explicit answers at present: Is the construction of eco-industrial parks conducive to the green development of cities? What is the influencing mechanism of eco-industrial parks on the green development of cities? Furthermore, is there any heterogeneity in the influences of eco-industrial parks on the green development of different types of cities? Answering these questions is not only beneficial to deepening the understanding of eco-industrial park policies but also conducive to improvements in the green development of cities in China and the realization of the Sustainable Development Goals.

Most of the existing academic studies focus on discussing the influences that eco-industrial parks have on environmental quality and economic efficiency in parks. Most scholars hold a positive opinion: supporting green industrial clusters plays an important role in improving the green economic efficiency of cities. They believe that green industrial clusters are crucial to the green transformation and high-quality development of cities [22–25]. These studies can generally be divided into qualitative analyses and quantitative analyses. Qualitative analyses concentrate on the development mode, evaluation studies, and existing problems of eco-industrial parks. The term eco-industrial park generally refers to a new industrial organization form, which is designed and built in a specific region through administrative means according to the requirements of clean production, the concept of the circular economy, and the principles of industrial ecology [11,26,27]. Eco-industrial parks establish a coexistence network of companies by imitating natural ecosystems [28], increase the degree of industrial clusters, and form professional divisions of labor [29]. Hence, members of the cluster can access the nearby highly specialized parts, raw materials, machine equipment, commercial services, talents, and other input resources that are needed, thus lowering transaction costs [30–32]. Meanwhile, resource recycling can be realized in the cluster [33], which is conducive to improving the green efficiency of the economy [34]. However, cities lack experience in the selective introduction of projects, the establishment of a cluster support network, innovation environment, and cluster culture when promoting the construction of eco-industrial parks [35]. Furthermore, the internal ecological industrial cluster is not formed by an independent mechanism of industrial clusters and internal correlations between industries [36]. Instead, it is a spatial agglomeration of enterprises, which are attracted by governments at all levels, through taxation, land, and other preferential policies. Eco-industrial parks have similar industrial structures and lack an explicit industrial orientation and industrial division of labor, making it difficult for ecological clusters to have substantial green effects [37–39]. The quantitative analysis focuses on a performance evaluation of national eco-industrial parks and their influence on economic development. The performance evaluation of eco-industrial parks mainly compares the impact of these effects under different economic and environmental conditions through the life-cycle theory [40–42]. For example, Li et al. took steel manufacturing, paper, pulp, and petrochemical industries as examples, and proved that an efficient green supply chain of waterfront industrial parks can have significant economic and environmental benefits [43]. In addition, some scholars focus on the pollution reduction function of eco-industrial parks and discuss the environmental performance of eco-industrial parks from the perspective of pollution reduction. For example, Tian et al. took the project of China National Eco-Industrial Demonstration Park as an example to empirically verify that the construction of eco-industrial parks has a significant inhibitory effect on the total amount and intensity of sulfur dioxide and other pollutants discharged in the park [28]. Liu et al. believe that optimizing energy structure and improving energy efficiency can significantly reduce greenhouse gas emissions in the Beijing economic and technological development area [27]. Taking 106 eco-industrial parks in China as examples, Guo et al. conducted an

ecological performance evaluation of eco-industrial parks on the basis of a large amount of research data and found that the construction of eco-industrial parks is conducive to the reduction in pollutant emissions [44]. In a quantitative analysis of the impact that national eco-industrial parks have on economic development, Cao et al. empirically tested the contribution of eco-industrial parks to the high-quality development of the regional economy from a macro-perspective through the difference–difference model [45]. Kim adopted the input-output analysis method and found that the construction of ecological industrial zones is conducive to green economic growth from the perspective of the supply chain [46].

First of all, in terms of the measurement of the green development level, the traditional accounting of the green development level takes capital and labor as input factors, while energy consumption and environmental pollution are not included in the investigation system. Therefore, it is difficult to accurately describe the level of social green development [47–49]. In order to solve this problem, indicators such as energy consumption and environmental pollution are gradually incorporated into the accounting system for the level of green development [7,50]. The specific measurement methods mainly include the parametric method and non-parametric method: the parametric method can be subdivided into the Solow residual method and stochastic frontier analysis (SFA); the non-parametric method is mainly based on data envelopment analysis (DEA) and is usually combined with the index method. Commonly used indexes include the Malmquist index, the Luenberger index, and the Malmquist–Luenberger (ML) index [51–53]. Secondly, in terms of the influencing factors of green development, the current research perspective is quite rich, covering digital finance [54], foreign direct investment [55], industrial collaborative agglomeration, green innovation [12], financial agglomeration [56], smart city construction and other economic activities, economic phenomena, and government policies [20]. For example, Lin et al. found that the distortion of the factor market leads to a reduction in regional exports and foreign direct investment, which will have a negative impact on the level of regional green development [57]. Using the spatial Durbin model, Wang et al. verified that green technology innovation has a significant positive impact on the green development of cities, but has a significant negative impact on the green development of neighboring regions [58]. Using panel data from 60 countries from 2008 to 2018, Dong et al. empirically tested the impact of the digital economy on carbon emissions and found that the development of the digital economy significantly reduced carbon emission intensity, but promoted the increase in per capita carbon emissions [59].

Through this literature review, it can be found that although the existing literature on eco-industrial parks and urban green development has been studied from different angles and different levels, which provides a certain theoretical basis and experience for the further study of this paper, there are three deficiencies in general. Firstly, most studies concerning the relationship between eco-industrial parks and the green development of cities are theoretical analyses, but there is a lack of empirical tests based on empirical data. This not only causes incompleteness of studies but also brings errors in analysis results. Secondly, there is a lack of literature that systematically expounds on the internal logic of eco-industrial parks affecting urban green development. Thirdly, differences in policy effects brought by heterogeneous individuals are ignored. Therefore, this study may have the following three marginal contributions. Firstly, this paper enriches and expands the existing research on the impact of eco-industrial parks on urban green development. Based on the quasi-natural environment brought by the pilot policy of eco-industrial parks, this study intends to build a DID model to test the impact of eco-industrial parks on urban green development. Secondly, the paper examines the transmission mechanisms of policies from three aspects: "green innovation effect", "industrial structure optimization effect" and "environmental regulation effect", and carries out statistical tests. Thirdly, due to the influence of various factors such as history and geographical location, the performance of policy effects in cities with different locations and characteristics will also

show differentiation. Therefore, this paper further explores the heterogeneity of policy effects of eco-industrial parks in depth.

## 2. Theoretical Analysis and Research Hypothesis

Based on industrial ecological design, eco-industrial parks are conducive to decreasing pollutant emissions and realizing resource recycling, thus facilitating green and sustainable economic development. Eco-industrial parks are a scientific way to realize green industrial transformation and green economic development [60]. The construction of eco-industrial parks can train an ecological industrial network system by reasonably planning an industrial chain, and realize the regional sustainable green development of coordinated environmental protection and economic growth. The essence of eco-industrial parks is to promote the intensive use of resources and the environment, train a modern ecological industrial system through green industrial agglomeration, and realize the goals of resource recycling, product remanufacturing, and disposal by imitating biological ecosystems in nature. In this way, it achieves the ecological effect of gradient uses of materials and energy and resource sharing [61]. The impact of eco-industrial parks on urban green development is mainly realized through three channels: "green innovation effect", "industrial structure optimization effect", and "environmental regulation effect".

### 2.1. Green Innovation Effect

Improvements in green innovation level are an important way for eco-industrial parks to promote the green development of cities. The "Porter Hypothesis" proposes that. in the long term, strict and reasonable environmental regulation is beneficial for enterprises to improve their technological innovation level. As an important environmental regulation and measure, the construction of eco-industrial parks can urge enterprises to improve their production technological process and continuously stimulate green technological innovation, thus improving both the environmental performance and productivity of enterprises [62]. Specifically, eco-industrial park policy requires enterprises to reach a certain pollutant emission standard. There are three ways for enterprises to meet the pollutant emission standard: (1) decreasing production capacity; (2) buying controlling equipment; (3) technological innovation. In the short term, most enterprises prefer to decrease their production capacity and buy controlling equipment [63]. However, in the long term, technological innovation is the trend [64]. Enterprises can decrease the increases in costs caused by environmental regulations by improving production technologies. In other words, the "innovation compensation effect" offsets the "pollutant emission cost" [65]. With the improvement in enterprises' green innovation level, technological reform empowers the traditional industrial chain, accelerates the greening, intelligence, and recycling of production process, and continues to trigger the green transformation of various production organizations in terms of development strategies, product services and organizational systems, thus facilitating the green production system.

Based on the above theoretical analysis, this paper puts forward hypothesis H1 and hypothesis H2.

**H1.** *Eco-industrial parks will promote the green development of cities.*

**H2.** *Eco-industrial parks will promote the green development of cities by improving the regional green innovation level.*

### 2.2. Industrial Structure Optimization Effect

Industrial structure optimization is another important way for eco-industrial parks to promote the green development of cities. In the past four decades, China's economy has experienced high-speed development, which mainly relies on the high consumption and high input of energy. As the marginal effect of the past economic growth, driven by element and resource inputs, gradually declines, it is urgent for China to change the past "extensive economic growth mode" and change from the original investment-driving and

element-driving modes to innovation-driving. Eco-industrial parks emphasize resource recycling and green sustainable development modes, which are violated by the high material consumption, high energy consumption, and high pollution of traditional industries. This forces the local government to transform from their original traditional industries to high-tech industries [66]. At the same time, the construction of eco-industrial parks is beneficial to the formation of industrial clusters and generates a constellation effect that can lower costs for the flowing and sharing of innovation elements in the region, strengthen the optimal configuration of innovation elements in the region, and improve regional innovation levels. With the improvement in innovation level, technological changes are brought about, and new technologies can effectively drive traditional industrial upgrading through penetration, overturn, and reconstruction, which is attributed to their high added values, high correlation, and high permeability [67]. With the continuous advances in industrial structure, the pollution problems caused by traditional industrial structures will gradually be relieved and, finally, the green sustainable development of cities will be realized [68].

Based on the above theoretical analysis, this paper puts forward hypothesis H3.

**H3.** *Eco-industrial parks will promote the green development of cities by optimizing industrial structures.*

### 2.3. Environmental Regulation Effect

The strengthening of environmental regulation is the third important way for eco-industrial parks to promote urban green development. Eco-industrial demonstration parks require key pollutants to be discharged up to standards and key enterprises to implement cleaner production. Therefore, eco-industrial demonstration parks play a strong role in environmental regulation, which, to some extent, will place pressure on the operating profits of polluters and cause them to face higher compliance costs over a certain period of time [69]. In order to avoid the adverse effects brought by environmental regulations, polluters have the following choices: (1) Production departments or high-pollution production links can be moved from areas with strong environmental regulations to areas with weak environmental regulations, so as to evade environmental regulations and transfer environmental pressure; (2) The main polluters could actively explore green innovation and clean production, aiming to improve their green competitiveness and green development efficiency by introducing green technology and green energy through source control, and establish a new mode of green sustainable development that is friendly to the environment. As green development strategies are the future trend, the construction of ecological industrial parks will become the development mainstream and national environmental regulations will be further strengthened in the future, the only dominant strategy is to promote green transformation through green innovation, which will further promote the green development of cities [70,71]. In addition, eco-industrial demonstration parks require "industrial ecology", which aims to improve the efficiency of resource and energy use, reduce pollutant emissions and solve pollution problems at the source of production by deliberately imitating the natural ecosystem. Therefore, the stronger the regulations, the more significant the effect of green development.

Based on the above theoretical analysis, this paper puts forward hypothesis H4.

**H4.** *Eco-industrial parks will promote urban green development by strengthening environmental regulations.*

## 3. Research Design

### 3.1. Model Design

The DID method is a widely accepted tool for a quantitative assessment of the effects of policies. Unlike traditional static comparison methods, the DID method does not directly measure the change in the mean of the sample before and after the policy but instead adds two dummy variables of policy and time and their interaction terms to the regression equation [72]. On the one hand, the DID method can capitalize on the homogeneity of the

explanatory variables by effectively controlling for the effects of unobservable individual heterogeneity on the explanatory variables. On the other hand, the DID method allows for unbiased estimation of policy effects without the unpredictable effects of time-varying aggregate factors. According to the Notices of Releasing the List of National Ecological Industrial Demonstration Parks published by the Ministry of Environmental Protection, MOST, and Ministry of Commerce, there were 48 officially named ecological industrial demonstration parks in China by the end of 2016, including 3 founded in 2008, 8 founded in 2010, 3 founded in 2011, 3 founded in 2012, 5 founded in 2013, 11 founded in 2014, 3 founded in 2015 and 12 founded in 2016. These ecological industrial demonstration parks were used as reform pilots to explore industrial ecologicalization. These provide us with a good quasi-natural experiment. This paper refers to the multi-phase DID model constructed by Beck et al. (2010) [73], Wang (2023) [74], and other scholars in their research to evaluate the empirical strategy of policy effects, and attempts to construct a multi-phase DID model to conduct an empirical test on the policy effects of eco-industrial park construction on urban green development. The specific formula is as follows:

$$\text{Gtfp}_{i,t} = \alpha_0 + \alpha_1 \text{treatpost}_{i,t} + \sum \beta_{i,t} \text{Controls}_{i,t} + \gamma \text{Year}_t + \mu \text{Region}_i + \varepsilon_{i,t} \qquad (1)$$

where i and t refer to the city i and year t. Gtfp is the explained variable, and it denotes the green development level of cities. The interaction term treatpost is a policy dummy variable and expresses whether the city is enlisted into the pilot cities of eco-industrial park construction. It is used to measure policy impact and is the core explanatory variable in this study. It should be noted that the above 48 eco-industrial parks may not distribute in different cities. In other words, some cities may have many eco-industrial parks. Hence, cities with only one eco-industrial park were defined according to the year that the eco-industrial park was established in this study. They were valued at 0 before and 1 after. For cities with many eco-industrial parks, only the year the first eco-industrial park was established was used. They were valued at 0 before the year of establishing the first eco-industrial park, but 1 after. Controls is a control variable. Year denotes the time-fixed effect. Region refers to the individual fixed effect of cities. The $\alpha, \beta, \gamma, \mu$ represent the regression coefficients of equations. The $\alpha_1$ reflects the policy effect in this study and its signs (positive or negative) objectively reflect whether eco-industrial parks promote or inhibit the green development of cities. The $\varepsilon$ is the random error term.

*3.2. Variable Declaration*

3.2.1. Explained Variables

The green development level (Gtfp) of cities was measured by the green total factor productivity in this study. Since data Envelopment Analysis (DEA) needs no priori functional forms and distributional hypothesis, and it can enlist energy input and pollutant emissions into the analysis framework simultaneously, it overcomes many of the shortages of traditional methods to effectively estimate total factor productivity. Recently, DEA has been widely applied to estimations of green total factor productivity [75]. To overcome radial and angle measurement deviations in the traditional DEA model, Tone (2010) [76] proposed an SBM model which is based on the slack variable and considers unexpected output. On this basis, an unexpected output SBM model under constant returns to scale was constructed with references to the empirical strategies of Wang et al. (2022) [77], which was used to estimate green total factor productivity with MaxDEA7.0 software through the Global Malmquist–Luenberger (GML) index, using 2004 as the base period. The theoretical description of this method has been thoroughly introduced [78] and is not further detailed in this study due to the limited space.

The practices of Jiang et al. (2021) [79] and Xin et al. (2019) [80] were referred to during the selection of input indexes, expected output index, and unexpected output index. Input indexes were material capital, labor capital, and resource losses. Economic development was chosen as the expected output, while pollutant emissions due to economic development

were chosen as the unexpected output. On this basis, an index system to measure the green development level of cities was established. Indexes are introduced in Table 1.

**Table 1.** Selection of indexes in the measurement system of green development level of cities in China.

| Level-1 Indexes | Level-2 Indexes | Level-3 Indexes |
|---|---|---|
| Input indexes | Material Capital | Capital stock/10,000 yuan |
| | Labor Capital | Quantity of employment at the end of a year/people |
| | Resource Losses | Total electricity consumption of a city/Wh |
| Expected output index | Economic Development | GDP at constant price/yuan |
| Unexpected output index | Pollutant Emission | Industrial wastewater emission/tons |
| | | Industrial fumes emission/tons |
| | | Industrial $SO_2$ emission/tons |
| | | PM2.5 emission-annual average PM2.5 concentration ($\mu g/m^3$) |

The temporal and spatial green development levels of cities (2004, 2009, 2013, and 2019) in China are shown in Figures 1 and 2, respectively. Results show that, from 2004 to 2019, the green development level of cities in China presented a rising trend. With respect to locations, cities in eastern China had a higher green development level than cities in central and western China. The green development level of cities in northern China increased quickly during 2004–2019.

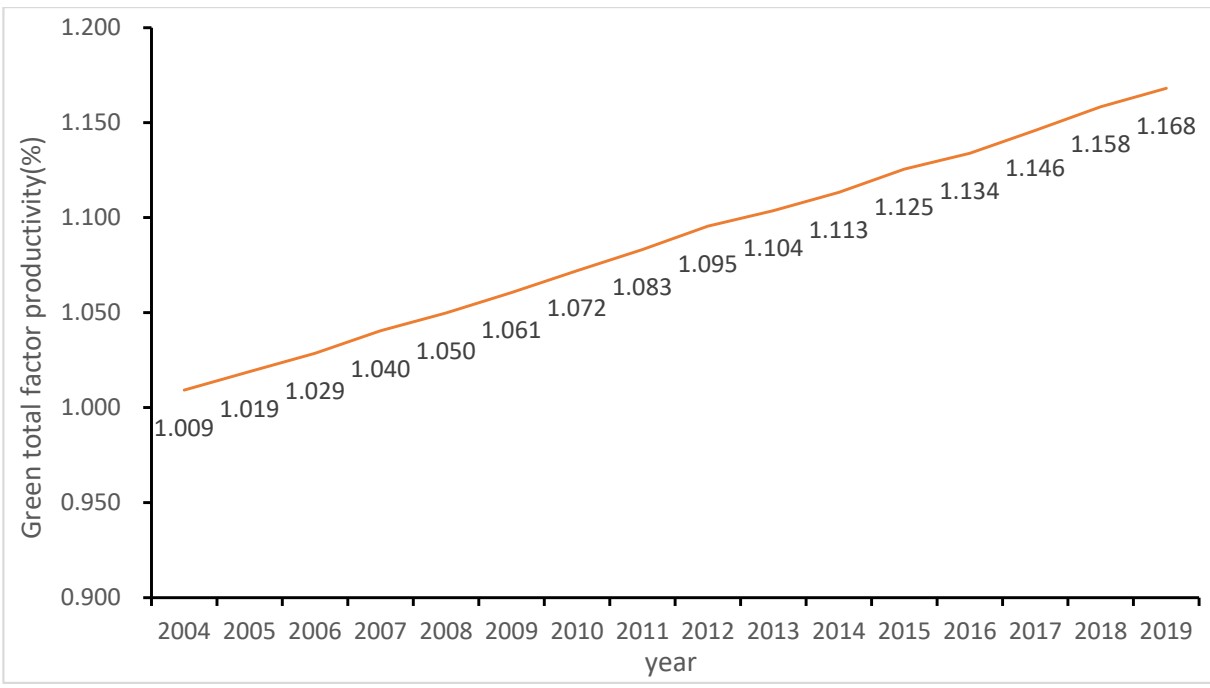

**Figure 1.** The mean green development level of cities in China during 2004–2019.

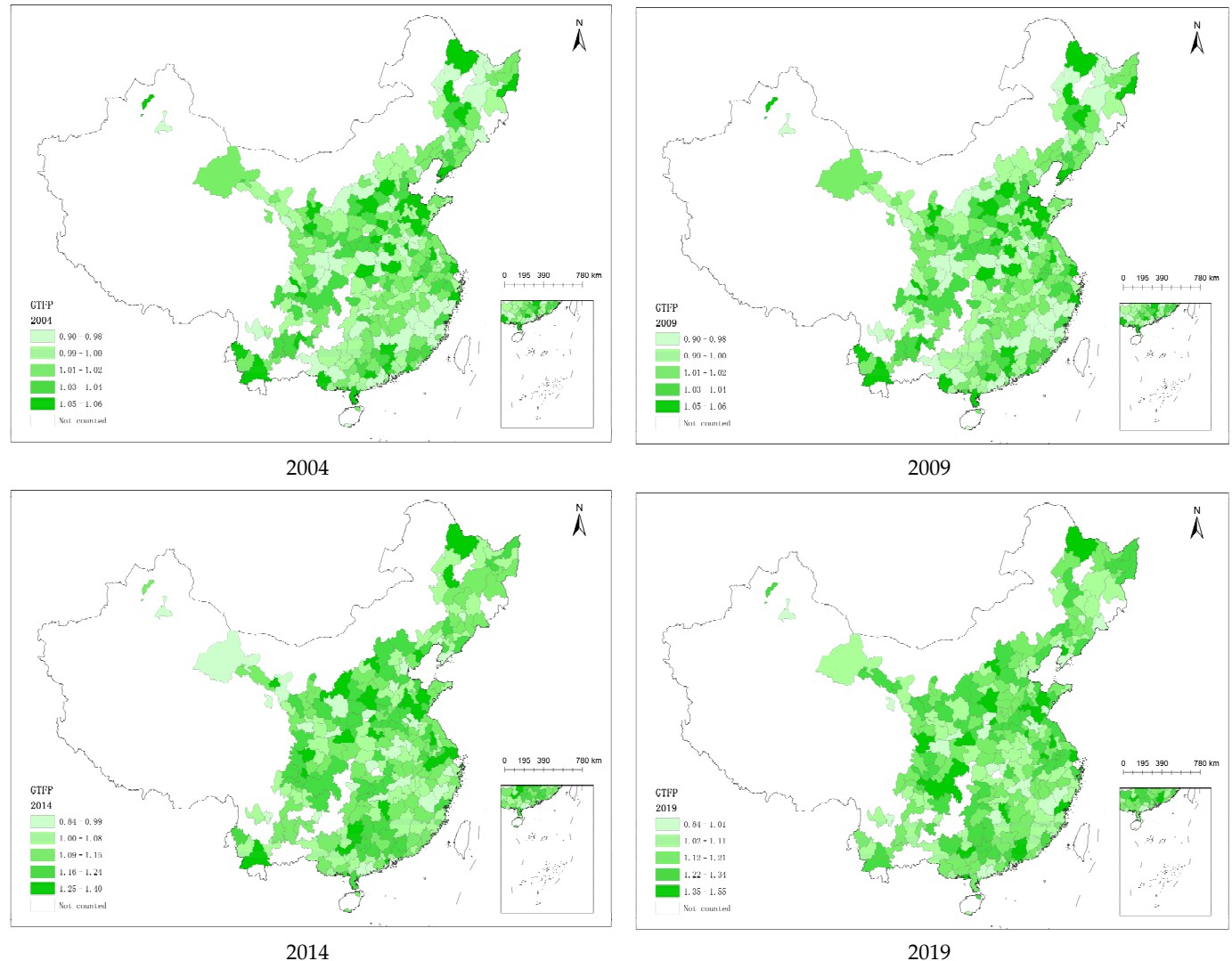

**Figure 2.** Green development levels of cities in 2004, 2009, 2013, and 2019.

### 3.2.2. Core Explanatory Variable

The policy dummy variable (treatpost) is the interaction term for the group dummy variable (Du) and the time dummy variable (Dt). Specifically, if a city is listed as a pilot construction city of an ecological industrial demonstration park, the grouping dummy variable (Du) is 1; otherwise, it is 0. At the same time, if the policy of the pilot construction of ecological industrial demonstration parks is proposed in a certain year, the year and subsequent years of the pilot policy are defined as 1 (Dt = 1); otherwise, the definition is 0 (Dt = 0). Treatpost = Du × Dt: when its value is 1, we consider the city to be undergoing a pilot reform; when its value is 0, we consider the city to be undergoing no pilot reform.

### 3.2.3. Mechanism Variable

Urban green innovation (greinv_grant, greinv_app). Referring to the empirical strategy of Yuan et al. (2022) [81] and Lin et al. [82], this paper selected the number of urban green invention licenses and the number of urban green invention patent applications as indicators to measure the level of urban green innovation. The original data of this paper came from the China Intellectual Property Office.

Industrial structure advancing level (indadv_lev). With reference to the calculation formula of the industrial structure advancing index of Wang (2022) [83], the industrial structure advancing index is defined as follows: firstly, GDP was divided into three parts according to three industries, and proportions of the added value of each part in GDP was

used as a component in the spatial vector. In this way, a group of 3D vectors $X_0 = (x_{1,0}, x_{2,0}, x_{3,0})$ was formed. Secondly, the included angles ($\theta_1$, $\theta_2$, and $\theta_3$) of $X_0$ with vectors of industrial arrangement, from low level to high level ($X_1 = (1, 0, 0)$, $X_2 = (0, 1, 0)$, $X_3 = (0, 0, 1)$)) were calculated according to:

$$\theta_j = \arccos\left(\frac{\sum_{i=1}^{3}\left(x_{i,j} \times x_{i,0}\right)}{\left(\sqrt{\sum_{i=1}^{3}\left(x_{i,j}^2\right)} \times \sqrt{\sum_{i=1}^{3}\left(x_{i,0}^2\right)}\right)}\right) j = 1, 2, 3 \tag{2}$$

Furthermore, the calculation formula of indadv_lev is defined as:

$$\text{indadv\_lev} = \sum_{t=1}^{3}\sum_{j=1}^{3} t \times \theta_j \tag{3}$$

In the above equation, indadv_lev is the industrial structure advancing index and the higher value indicates that the industrial structural updating is quicker.

The intensity of environmental regulation (env_reg). With reference to the practices of Yin et al. (2022) [84], this paper constructs a comprehensive index from three dimensions of environmental regulation level, environmental regulation pressure, and environmental regulation investment, and comprehensively measures the intensity of environmental regulation by using entropy method and range method. The specific steps are as follows:

$$\text{Positive indicators}: x_{ijt} = \frac{a_{ijt} - \min\left(a_{ijt}\right)}{\max\left(a_{ijt}\right) - \min\left(a_{ijt}\right)} \tag{4}$$

$$\text{Negative indicators}: x_{ijt} = \frac{\max\left(a_{ijt}\right) - a_{ijt}}{\max\left(a_{ijt}\right) - \min\left(a_{ijt}\right)} \tag{5}$$

where, $a_{ijt}$ represents the initial data of the j indicator in the i th city, in the t year; $x_{ijt}$ represents the value $a_{ijt}$ after standardization; $\max(a_{ijt})$ represents the maximum value corresponding to the j indicator,; and $\min(a_{ijt})$ represents the minimum value corresponding to the j indicator. Further, calculate the weight of the j index in the i city in the t year:

$$p_{ijt} = \frac{x_{ijt}}{\sum_{i=1}^{N}\sum_{t=1}^{T} x_{ijt}} \tag{6}$$

where, n = 276 is the number of sample cities, and T = 16 is the number of years.

Calculate the information entropy and redundancy of the j index:

$$e_j = \frac{1}{\ln NT}\sum_{i=1}^{N}\sum_{t=1}^{T}\left(p_{ijt} \times \ln p_{ijt}\right) \tag{7}$$

$$d_j = 1 - e_j \tag{8}$$

$$e_j \in [0, 1] \tag{9}$$

Calculate the weight of the j index according to the redundancy of information entropy:

$$w_j = \frac{d_j}{\sum_{j=1}^{m} d_j} \tag{10}$$

where m is the total number of indicators.

Finally, based on standardized indicators and measured index weights, environmental regulation intensity (env_reg) is calculated using multiple linear weighting function method:

$$\text{env\_reg}_{it} = \sum_{j=1}^{m} w_j \times x_{ijt} \tag{11}$$

The specific measurement index system is selected as shown in Table 2.

**Table 2.** Selection of indicators for the measurement system of environmental regulation intensity in China's prefecture-level cities.

| Target Layer | First-Order Index | | Secondary Index | | Index Layer |
|---|---|---|---|---|---|
| | **Subsystem** | **Weight** | **Basic Index** | **Weight** | |
| Environmental regulation strength | Environmental regulation level | 0.5017 | Park green area per capita (m$^2$) | 0.1022 | + |
| | | | Length of the urban drainage pipeline (km) | 0.0905 | + |
| | | | Domestic waste treatment rate (%) | 0.1034 | + |
| | | | Sewage treatment rate (%) | 0.1028 | + |
| | | | Industrial solid waste utilization rate (%) | 0.1028 | + |
| | Environmental regulatory pressure | 0.3950 | Industrial sulfur dioxide emissions (tons) | 0.0852 | − |
| | | | Industrial wastewater discharge (10,000 tons) | 0.1028 | − |
| | | | Industrial smoke (dust) emissions (tons) | 0.1036 | − |
| | | | Industrial nitrogen oxide emissions (tons) | 0.1034 | − |
| | Environmental regulation investment | 0.1033 | Per capita financial expenditure on energy conservation and environmental protection (yuan) | 0.1033 | + |

### 3.2.4. Control Variables

Many factors influence the green development level of cities. To decrease interference from other factors regarding the research problem, some control variables were chosen to strengthen the objectivity of evaluations of eco-industrial park construction policies. Moreover, based on existing studies and data accessibility, control variables were chosen from the perspectives of urban economic development level, urban social environment, and urban government management. The control variables include economic development level (pgdp), industrial structure (lnsec_ind) [85], scientific and technological level (scie), fixed asset investment level (fix_pro), financial development level (FIN) [56], government intervention ability (gov) and average wage (lnavgwag_emp). The measurement standards of the different variables are listed in Table 3.

**Table 3.** Names, signs, and definitions of variables.

| Name of Variables | Signs | Definitions of Variables |
|---|---|---|
| Green total factor productivity | Gtfp | —— |
| Policy dummy variable | treatpost | 1 for pilot cities of the eco-industrial park; otherwise, 0 |
| Urban green innovation | greinv_grant | Number of granted patents for urban green inventions |
| | greinv_app | Number of patent applications for urban green inventions |
| Industrial structure advancing | indadv_lev | $indadv\_lev = \sum\limits_{t=1}^{3} \sum\limits_{j=1}^{3} t \times \theta_j$ |
| Environmental regulation strength | env_reg | Through the environmental regulation grade, environmental regulation pressure, and environmental regulation investment to construct a comprehensive index measure |
| Economic development level | pgdp | Per capita GDP of a region |
| Industrial structure | lnsec_ind | The proportion of the added value of the secondary industry in GDP in the region |
| Scientific and technological level | scie | The proportion of scientific support in GDP |
| Fixed asset investment level | fix_pro | The proportion of fixed asset investment in GDP |
| Financial development level | FIN | The proportion of loan balance of financial institutions at the end of the year in GDP |
| Government intervention ability | gov | The proportion of expenses within general budgets of local finance in GDP |
| Average wage | lnavgwag_emp | Average wages of workers |

### 3.3. Data Source

The paper according to the principles of data accessibility, objectivity, and authenticity, all cities in Tibet and some cities in Qinghai, as well as cities with severe data missing, were eliminated. Finally, panel data of 276 prefecture-level cities in China spanning from 2004–2019 were chosen by paper. In this paper, the index of missing data is treated with linear interpolation. To avoid estimation errors of the model due to data outliers, a bilateral 2% winsorization was performed on the original data by paper. The original data for the estimation of green total factor productivity came from the past China City Statistical Yearbook, China Energy Statistical Yearbook, China Environmental Statistical Yearbook, and National Statistics Bureau. Data to measure urban green innovation level came from the China Intellectual Property Office. The rest original data came from the past China City Statistical Yearbook, China Energy Statistical Yearbook, and China Environmental Statistical Yearbook. Data collection and review were all completed manually by the research team.

### 3.4. Statistical Description of Variables

Before the reference regression, statistical descriptions of major variables are needed. The statistical description results of major variables are shown in Table 4. In view of the results, the mean Gtfp is 1.088. The minimum and maximum are 0.792 and 1.548, showing a difference of nearly two times. This indicates that there is a difference in green development levels among cities within the setting time interval. Regarding the number of granted urban green invention patents, the maximum and

minimum results for urban green innovation are 16.661 and 0.001, and the mean is only 0.129. This reflects that the overall green innovation level of research objects is generally low, and some cities are ahead of other cities. The mean of indadv_lev is 6.441 and the median is 6.421, indicating that the industrial structure advancing index of more than 50% of cities is higher than the mean. The statistical description results of the rest control variables are relatively close to the results of existing studies [85] and they are not further detailed in the present study.

**Table 4.** Statistical description of variables.

| Variable | Obs. | Mean | Std. Dev. | Min | Max | Variable |
|----------|------|------|-----------|-----|-----|----------|
| Gtfp | 4416 | 1.088 | 0.103 | 0.792 | 1.072 | 1.548 |
| treatpost | 4416 | 0.046 | 0.210 | 0.000 | 0.000 | 1.000 |
| greinv_grant | 4400 | 0.546 | 1.197 | 0.001 | 0.129 | 16.661 |
| greinv_app | 4416 | 1.144 | 2.284 | 0.009 | 0.288 | 26.150 |
| indadv_lev | 4416 | 6.441 | 0.350 | 5.429 | 6.421 | 7.836 |
| env_reg | 4416 | 0.607 | 0.110 | 0.099 | 0.619 | 0.884 |
| pgdp | 4416 | 3.813 | 2.930 | 0.449 | 3.014 | 14.928 |
| lnsec_ind | 4416 | 3.830 | 0.263 | 2.368 | 3.868 | 4.495 |
| scie | 4416 | 0.195 | 0.189 | 0.006 | 0.142 | 1.075 |
| fix_pro | 4416 | 1.036 | 1.167 | 0.038 | 0.662 | 6.713 |
| FIN | 4416 | 0.874 | 0.506 | 0.270 | 0.714 | 2.958 |
| gov | 4416 | 0.171 | 0.096 | 0.040 | 0.147 | 0.756 |
| lnavgwag_emp | 4416 | 10.447 | 0.586 | 8.509 | 10.526 | 12.062 |

## 4. Empirical Analysis

### 4.1. Reference Regression Results

In order to test the accuracy of hypothesis H1, the bidirectional fixed effect model is used in this paper to perform a regression estimation for Equation (1). Table 5 reports the empirical results of the relationship between eco-industrial parks and urban green development. Model (1) shows that, in addition to controlling individual effects and year effects, without adding any control variables, the implementation of pilot policies for eco-industrial parks can promote urban green development at a 1% significance level. From Model (2) to Model (4), control variables of the urban economic level, urban social environment, and urban government management are added successively. The results show that treatpost is still significantly positive at the 1% level. Meanwhile, OLS regression results are listed for the convenience of comparison with bidirectional fixed effects. It can be seen that OLS regression results do not change the basic conclusions of the bidirectional fixed effect model. In sum, the implementation of the pilot policy of eco-industrial parks can significantly promote the urban green development of cities, with an increase of about 0.0279 units, which coincides with the conclusion of existing studies [60,63]. The eco-industrial park is a new type of industrial park based on the requirements of clean production, the concept of the circular economy, and the principles of industrial ecology, seeking the mutual promotion of economic and environmental benefits. It can transform pollutant emissions or by-products from different enterprises into recyclable and renewable substances by imitating the ecosystem, thus reducing pollution emissions and promoting urban green development. The pollution emissions of enterprises in the park must meet the standards stipulated by the government, which further regulate enterprises' pollution emissions so as to achieve green development. Thus, hypothesis H1 is verified.

**Table 5.** Reference regression.

| Name of Variables | Gtfp | | | | |
|---|---|---|---|---|---|
| | **Model (1)** | **Model (2)** | **Model (3)** | **Model (4)** | **Model (5)** |
| treatpost | 0.0208 *** (0.0054) | 0.0274 *** (0.0057) | 0.0282 *** (0.0057) | 0.0279 *** (0.0057) | 0.0399 *** (0.0073) |
| pgdp | | −0.0030 *** (0.0009) | −0.0023 ** (0.0009) | −0.0029 *** (0.0010) | 0.0018 ** (0.0008) |
| lnsec_ind | | 0.0006 (0.0039) | 0.0004 (0.0039) | 0.0023 (0.0040) | −0.0197 *** (0.0054) |
| scie | | | −0.0207 *** (0.0079) | −0.0210 *** (0.0080) | −0.0349 *** (0.0096) |
| fix_pro | | | 0.0009 (0.0008) | 0.0015 * (0.0009) | −0.0018 (0.0013) |
| FIN | | | | 0.0104 ** (0.0046) | −0.0211 *** (0.0031) |
| gov | | | | −0.0590 ** (0.0265) | 0.0725 *** (0.0193) |
| lnavgwag_emp | | | | −0.0068 (0.0053) | 0.0766 *** (0.0038) |
| constant | 1.0093 *** (0.0032) | 1.0111 *** (0.0151) | 1.0109 *** (0.0151) | 1.0670 *** (0.0484) | 0.3688 *** (0.0397) |
| City | Yes | Yes | Yes | Yes | No |
| Time | Yes | Yes | Yes | Yes | No |
| $R^2$ | 0.4846 | 0.4860 | 0.4870 | 0.4883 | 0.2123 |
| *n* | 4416 | 4416 | 4416 | 4416 | 4416 |

Note: Values in parentheses are standard deviations; *, **, and *** indicate significance at the 10%, 5%, and 1% significance levels, respectively.

According to the control variables listed in Model (5), the level of economic development (pgdp) is negative at the significance level of 1%, indicating that China's economic development mainly relies on traditional factors and has not fully realized the economic entity transformation. According to the environmental Kuznets theory, there is a U-shaped relationship between economic growth and environmental quality, which also shows that the current level of China's economic development has not reached the inflection point of the environmental Kuznets curve [86]. Industrial structure (lnsec_ind) is positively correlated with the level of urban green development, but the correlation is not significant, which indicates that, in recent years, China has achieved certain results in the adjustment of the internal structure of the secondary industry, not only by strictly controlling the growth rate of high-energy-consuming and high-emission industries but also by increasing the proportion of low-energy-consuming and low-emission industries. Science and technology level (scie) is negatively correlated with green development, and this is significant at the 1% level. A possible reason for this is that, at present, China's science and technology investment is mostly concentrated on non-green development, emphasizing economic development while ignoring environmental governance. With the improvements in science and technology level, the economic development level will continue to rise, but economic development at the early stage mostly comes at the cost of environmental pollution. As a result, scientific and technological capital investment increases, and environmental pollution increases. However, in the long term, improvements in the scientific and technological level will bring about improvements in the environment, which also shows that the Chinese government should properly guide the scientific and technological funds at the green development level when increasing financial investments in science and technology and advocating scientific and technological innovation, to allow for the role of science and technology in supporting green development [87]. Fixed-asset investment (fix_pro)

can promote urban green development at the level of 10%, which indicates that the construction of a reasonable infrastructure is the premise of urban green development, and good urban infrastructure construction can attract more talent and capital inflows, thus forming a factor agglomeration effect and creating a possible space for urban green development [88]. The financial development level (FIN) promotes the green development of the city at the significance level of 5%, indicating that the higher the financial development level of the city, the more loanable enterprise funds exist, and there are more possible loans for green innovation and green production. Government intervention capacity (gov) inhibits green development in cities at the 5% level for the following possible reasons: With further government intervention in the market, more serious phenomena, such as resource allocation distortion and market resource crowding, will be caused, leading to the limited possibility of the market obtaining resources for green development. However, most local government officials in China attach importance to financial support for economic development, ignoring the promotion of environmental governance and other work, for reasons such as political promotion. This brings about serious environmental pollution and other problems [57]. The average wage (lnavgwag_emp) is negatively correlated with urban green development, but this has not passed statistical tests. The level of average wage also indirectly indicates the level of urban economic development, and the relationship between the two is negative, which once again verifies the basic conclusion that "China's economic development still mainly relies on traditional factors".

### 4.2. Parallel Trend Test

The DID model states that the external regulating variables must be random. This means that the treatment and control groups in the samples selected for this paper must meet a prior "common trend." This is an important prerequisite for effective estimation based on DID model. Specifically, before the introduction of the eco-industrial park pilot policy, the level of green development of pilot cities and non-pilot cities should have the same change trend. If this assumption cannot be satisfied, the difference in the trend of urban green development level before the implementation of the policy may lead to estimation errors. According to the test results (Figure 3), the interaction coefficient of the policy before issuing and implementation is near 0 and it is not significant. This proves that the paper's selected sample has a good prior hypothesis. It can also be seen from the graph that after the introduction of the policy, the difference between the green development level of pilot cities and non-pilot cities has gradually intensified. To sum up, the samples selected in this paper can satisfy the parallel trend hypothesis well.

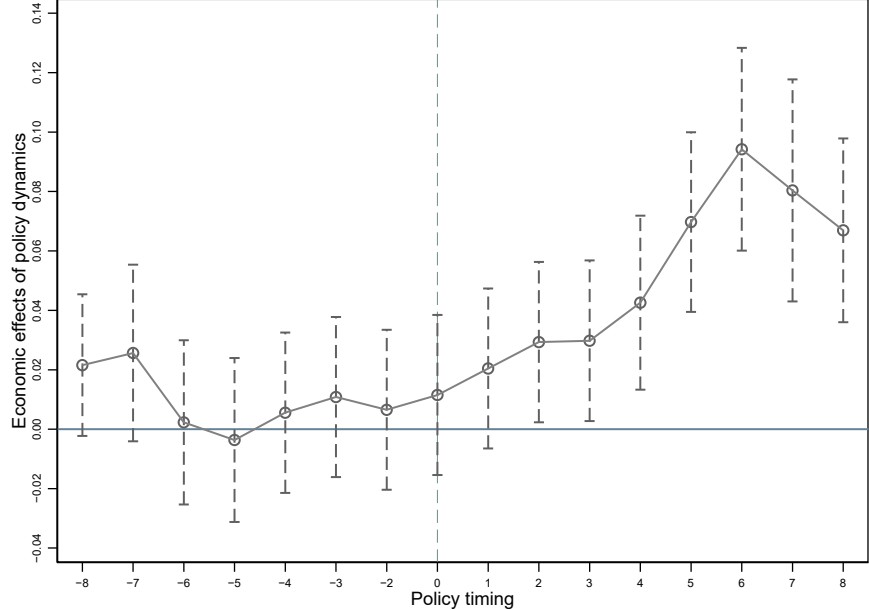

**Figure 3.** The dynamic effect of eco-industrial park pilot policy during 2004–2019.

### 4.3. Robustness Test

#### 4.3.1. Placebo Test

Urban fixed effect and year fixed effect were added to the reference regression model. Moreover, other relevant variables that may influence the green development of cities were controlled to relieve the estimation errors caused by missing variables as much as possible. However, unobservable factors may still affect the estimation results. With reference to existing studies [89], a placebo test was carried out by randomly choosing pilot cities in the "eco-industrial park" policy. Since the "pseudo"-treatment group is generated randomly through a numerical table, the "pseudo"-policy impact may not significantly influence improvements in the green development of cities in theory. In other words, the regression coefficients of samples for estimation after "pseudo"-treatment should be near 0 and present a normal distribution; otherwise, there is an error in the sample estimation.

Hence, the above random sampling process was repeated 500 times, and model estimation was implemented. A total of 500 regression coefficients were estimated. On this basis, the kernel density distribution pattern of estimation coefficients of the "pseudo"-policy impact variable was plotted (Figure 4). The results showed that, given a random treatment group and control group selection, the estimation coefficient observes normal distribution near 0 and the mean is close to 0. Moreover, the estimation value of the practical effects of the eco-industrial park pilot policy (0.0279) lies in the small probability interval in the kernel density distribution map. In other words, the promotion effect of eco-industrial parks on the green development level of cities is less likely to be attributed to unobservable factors. This further proves the reliability and robustness of the regression results in the present study.

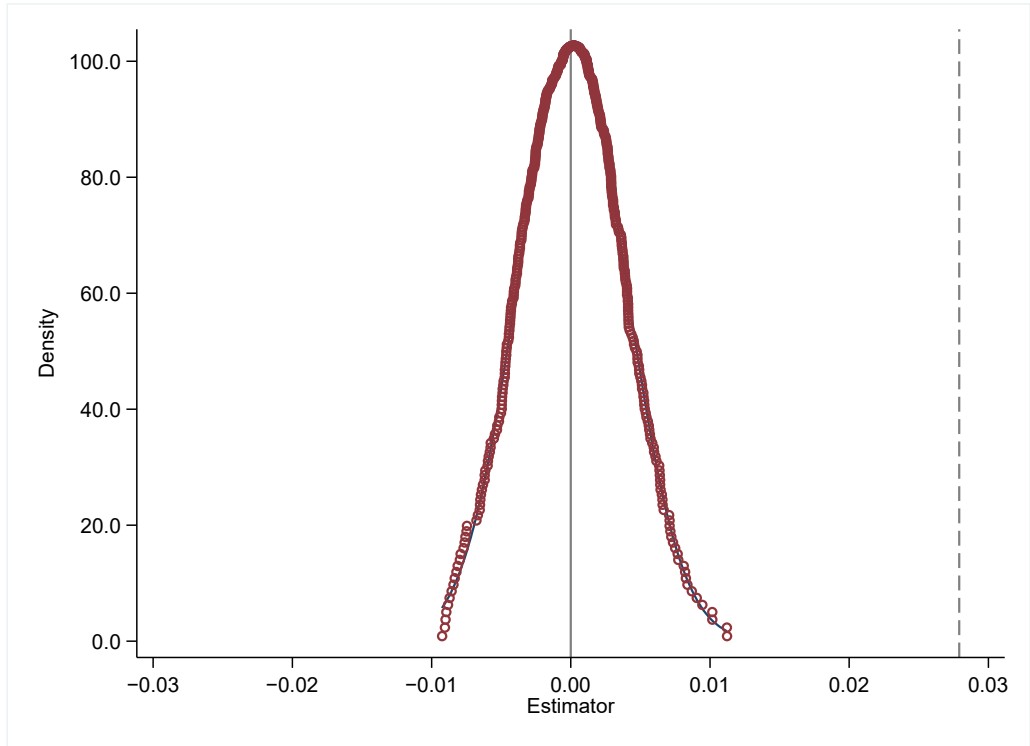

**Figure 4.** Placebo Test Results.

#### 4.3.2. PSM-DID Analysis Results

The DID model requires that the selection of the treatment group and the control group has no directionality. In this study, pilot cities of eco-industrial parks were selected completely randomly, which is very difficult to achieve in practical processes. This increases selective estimation errors to some extent. Secondly, when considering

regional differences and the heterogeneity of cities, estimation errors regarding policy effect are likely if there is a great difference in urban characteristics between the treatment group and the control group. To overcome this problem, this study used the propensity score-matching (PSM) method to find the city in the control group with the most similar characteristics for each pilot city, and then carry out another DID model estimation based on the matched samples (PSM-DID). Specifically, pgdp, pgdp, scie, fix_pro, FIN, gov, and lnavgwag_emp were chosen as matching variables and the Logit model was applied to estimate the propensity score. With references to the empirical strategy of Fu et al. (2021) [90], the 1:1 kernel matching method was used to obtain the matching samples. The PSM-DID regression analysis is shown in Model (1) of Table 6. Specifically, the treatpost coefficient is still positive at the 1% significance level. This reveals that, given the elimination of differences in urban characteristics, eco-industrial parks can significantly promote the green development of cities. This further supports the reference empirical conclusion of this study.

**Table 6.** Robustness test results.

| | Gtfp | | | |
|---|---|---|---|---|
| | **Model (1)** | **Model (2)** | **Model (3)** | **Model (4)** |
| **Name of Variables** | **PSM-DID** | **Eliminating Innovation Pilot Cities, Environmental Protection Tax Reform Policy** | **Eliminating Provincial Capitals and Municipalities** | **Log Control Variables** |
| treatpost | 0.0211 *** (0.0060) | 0.0244 *** (0.0059) | 0.0387 *** (0.0071) | 0.0283 *** (0.0058) |
| inno_year | | 0.0104 *** (0.0051) | | |
| taxref_year | | 0.0093 *** (0.0049) | | |
| pgdp | 0.0002 (0.0014) | −0.0034 *** (0.0011) | −0.0019 * (0.0011) | |
| lnsec_ind | 0.0019 (0.0069) | 0.0025 (0.0040) | 0.0027 (0.0043) | |
| scie | −0.0080 (0.0119) | −0.0218 *** (0.0080) | −0.0278 *** (0.0086) | |
| fix_pro | −0.0019 (0.0022) | 0.0016 * (0.0009) | 0.0018 ** (0.0009) | |
| FIN | 0.0044 (0.0077) | 0.0099 ** (0.0046) | 0.0166 *** (0.0050) | |
| gov | −0.1255 (0.0883) | −0.0536 ** (0.0266) | −0.0541 ** (0.0270) | |
| lnavgwag_emp | 0.0024 (0.0081) | −0.0069 (0.0053) | −0.0094 * (0.0055) | |
| $Pgdp_{t-1}$ | | | | −0.0041 *** (0.0011) |
| $lnsec\_ind_{t-1}$ | | | | 0.0028 (0.0042) |

**Table 6.** *Cont.*

| Name of Variables | Gtfp | | | |
| --- | --- | --- | --- | --- |
| | Model (1) | Model (2) | Model (3) | Model (4) |
| | PSM-DID | Eliminating Innovation Pilot Cities, Environmental Protection Tax Reform Policy | Eliminating Provincial Capitals and Municipalities | Log Control Variables |
| $Scie_{t-1}$ | | | | −0.0134 (0.0085) |
| $fix\_pro_{t-1}$ | | | | 0.0006 (0.0009) |
| $FIN_{t-1}$ | | | | 0.0103 ** (0.0049) |
| $Gov_{t-1}$ | | | | −0.0408 (0.0276) |
| $lnavgwag\_emp_{t-1}$ | | | | −0.0050 (0.0053) |
| constant | 1.0194 *** (0.0850) | 1.0665 *** (0.0484) | 1.0843 *** (0.0505) | 1.0568 *** (0.0487) |
| City | Yes | Yes | Yes | Yes |
| Time | Yes | Yes | Yes | Yes |
| $R^2$ | 0.3926 | 0.4892 | 0.5008 | 0.4723 |
| $n$ | 1862 | 4416 | 3952 | 4140 |

Note: Values in parentheses are standard deviations; *, **, and *** indicate significance at the 10%, 5%, and 1% significance levels, respectively.

### 4.3.3. Eliminating Other Interferences Influencing the Green Development Policy

Since the window period of this study occurred in the special stage of national economic and social transformation, there might be other policy impact variables that influence the green development level of cities, interfering with the estimation accuracy of policy effect. Hence, interferences caused by other relevant policies have to be eliminated. Based on a review of the literature and review of the contents of important reform measures in the same period, reform measures with potential influences on the green development level of cities mainly focus on the following two aspects.

(1) Technological innovation. In 2006, China put forward its strategy for constructing an innovative country. In 2008, Shenzhen was chosen as the first innovation pilot city, followed by seven successive pilots. By January 10th, 2022, a total of 103 innovation pilot cities were established. The construction of innovation pilot cities is conducive to improving regional innovation levels. New technologies are conducive to traditional industrial upgrading as they can penetrate traditional industries, develop green economic growth points in a new form, and promote improvements in the green development of cities [91].

(2) Environmental regulation. The Environmental Protection Tax Law of the People's Republic of China was officially implemented in 2018. As a command control environmental regulatory policy, environmental protection tax has more universal constraints against enterprises. Environmental tax forces enterprises to innovate technologies and update equipment by increasing the taxation standards of pollutant emission and increasing the cost burden for enterprises. It breaks the path dependence of the development of traditional pollution-intensive enterprises and thereby realizes the goal of green sustainable development [92].

Based on the above analysis, the dummy variables of innovation pilot city policy (inno_year) and environmental protection tax reform policy (taxref_year) were input into the basic regression equation as control variables to test the policy effect of eco-industrial parks on the green development level of cities while the interferences of other policies are eliminated. Regression results are shown in Model (2) in Table 6. The dummy variable of the eco-industrial park policy (treatpost) is still significantly positive at the 1% level. This proves that the empirical test conclusion is still robust after the interferences of other policies are eliminated.

### 4.3.4. Eliminating Provincial Capitals and Municipalities

In China, provincial capitals and municipalities often develop well and possess richer resources due to their special locations; therefore, they have a great advantage in industrial restructuring, technological innovation, and policy support. This might influence the results [93]. As a result, provincial capitals and municipalities are directly eliminated, and only ordinary prefecture-level city samples are used in the regression. The regression results are shown in Model (3) in Table 6. The result is still significantly positive at the 1% level, indicating that the reference regression results are very robust.

### 4.3.5. Log Control Variables

To solve the endogenous problem of control variables, that is, the reverse causality between the explained variables and control variables, log control variables were included in the basic model for secondary DID model estimation in this study. The results are shown in Model (4) in Table 6. After comparison with Model (4) in Table 5 and Model (4) in Table 6, the regression coefficient of the policy dummy variable (treatpost) is significantly positive at the 1% level, indicating that the conclusion is relatively reliable: eco-industrial parks can improve the green development level of cities.

### 4.4. Action Mechanism Analysis

Based on previous empirical results, the construction of eco-industrial parks can promote significant improvements in the green development level of cities. Therefore, the action mechanism and pathway of eco-industrial parks are worthy of further study. Based on a previous theoretical analysis, the following hypothesis is proposed: eco-industrial parks influence the green development level of cities through the "technological innovation effect", "industrial structure optimization effect" and "environmental regulation effect". This hypothesis is verified in this section. With reference to the practices of Shi et al. [94], the following transmission effect model was constructed to recognize the action mechanism of eco-industrial parks on the green development of cities.

$$\text{Mid}_{i,t} = \alpha_0 + \alpha_1 \text{treatpost}_{i,t} + \sum \beta_{i,t} \text{Controls}_{i,t} + \gamma \text{Year}_t + \mu \text{Region}_i + \varepsilon_{i,t} \quad (12)$$

$$\text{Gtfp}_{i,t} = \alpha_0 + \alpha_1 \text{Mid}_{i,t} + \sum \beta_{i,t} \text{Controls}_{i,t} + \gamma \text{Year}_t + \mu \text{Region}_i + \varepsilon_{i,t} \quad (13)$$

where $\text{Mid}_{it}$ is the mechanism variable, which refers to regional green innovation, industrial structural advances, and environmental regulation intensity. The remaining variables are defined in the same way and they are not further introduced here.

The test results of the transmission effect are shown in Table 7. The regression coefficients of the policy dummy variable (treatpost) in Model (1) and Model (3) are significantly positive at the 1% level. In other words, eco-industrial parks can promote improvements in the regional green innovation level. In Model (2) and Model (4), the regression coefficients of indexes to measure urban green innovation level are significantly positive at the 5% and 1% levels. This implies that improving the regional innovation level is conducive to the green development of cities. Based on the above analysis, eco-industrial parks can promote the green development of cities by improving regional green innovation levels. In Model (5), the regression coefficient of the policy dummy variable (treatpost) is significantly positive under the 5% level, indicating that eco-industrial parks can optimize the industrial

structure. In Model (6), the regression coefficient of the industrial structure advancing index (indadv_lev) is significantly positive at the 5% level. This reflects that eco-industrial parks may promote the green development of cities by optimizing the industrial structure. Based on the above analysis, eco-industrial parks can promote improvements in the green development level of cities by optimizing the industrial structure. In addition, in Model (6), treatpost is significantly positive at the 5% level, while in Model (7), the regression result of the environmental regulation intensity proxy variable (env_reg) is also significantly positive at the 5% level, indicating that eco-industrial parks have the right to enhance the level of urban green development by strengthening environmental regulation intensity. Hypothesis H2, hypothesis H3, and hypothesis H4 of this paper have also been verified.

**Table 7.** Mechanism test results.

| Name of Variables | Regional Green Innovation | | | |
|---|---|---|---|---|
| | Model (1) greinv_grant | Model (2) *Gtfp* | Model (3) greinv_app | Model (4) *Gtfp* |
| treatpost | 0.2363 *** (0.0355) | | 0.5826 *** (0.0692) | |
| greinv_grant | | 0.0064 ** (0.0025) | | |
| greinv_app | | | | 0.0055 *** (0.0013) |
| pgdp | 0.0638 *** (0.0064) | −0.0020 * (0.0010) | 0.0920 *** (0.0125) | −0.0022 ** (0.0010) |
| lnsec_ind | −0.0090 (0.0252) | 0.0016 (0.0041) | −0.0574 (0.0489) | 0.0013 (0.0040) |
| scie | 0.4970 *** (0.0496) | −0.0218 *** (0.0081) | 0.6800 *** (0.0968) | −0.0231 *** (0.0081) |
| fix_pro | −0.0138 ** (0.0056) | 0.0015 * (0.0009) | −0.0159 (0.0109) | 0.0015 * (0.0009) |
| FIN | 0.0724 ** (0.0286) | 0.0111 ** (0.0046) | 0.1116 ** (0.0557) | 0.0104 ** (0.0046) |
| gov | 0.0786 (0.1648) | −0.0637 ** (0.0267) | 0.0233 (0.3205) | −0.0590 ** (0.0265) |
| lnavgwag_emp | 0.0566 * (0.0327) | −0.0076 (0.0053) | 0.0652 (0.0635) | −0.0070 (0.0053) |
| constant | −0.1852 (0.3015) | 1.0725 *** (0.0488) | 0.4446 (0.5853) | 1.0656 *** (0.0485) |
| City | Yes | Yes | Yes | Yes |
| Time | Yes | Yes | Yes | Yes |
| $R^2$ | 0.2087 | 0.4846 | 0.1148 | 0.4876 |
| *n* | 4400 | 4400 | 4416 | 4416 |

| Name of Variables | Industrial Structure Optimization Effect | | Environmental Regulation Effect | |
|---|---|---|---|---|
| | Model (5) indadv_lev | Model (6) *Gtfp* | Model (7) env_reg | Model (8) Gtfp |
| treatpost | 0.0192 ** (0.0087) | | 0.0098 ** (0.0048) | |
| indadv_lev | | 0.0217 ** (0.0103) | | |
| env_reg | | | | 0.0366 ** (0.0185) |
| pgdp | −0.0114 *** (0.0016) | −0.0013 (0.0010) | −0.0003 (0.0009) | −0.0015 (0.0010) |
| lnsec_ind | −0.0113 * (0.0062) | 0.0011 (0.0040) | −0.0054 (0.0034) | 0.0011 (0.0040) |

**Table 7.** *Cont.*

| Name of Variables | Industrial Structure Optimization Effect | | Environmental Regulation Effect | |
|---|---|---|---|---|
| | Model (5) indadv_lev | Model (6) *Gtfp* | Model (7) env_reg | Model (8) Gtfp |
| scie | 0.0326 *** (0.0122) | −0.0200 ** (0.0080) | −0.0244 *** (0.0068) | −0.0184 ** (0.0080) |
| fix_pro | −0.0002 (0.0014) | 0.0015 (0.0009) | −0.0011 (0.0008) | 0.0015 (0.0009) |
| FIN | 0.0271 *** (0.0070) | 0.0105 ** (0.0046) | −0.0031 (0.0039) | 0.0112 ** (0.0046) |
| gov | −0.1670 *** (0.0403) | −0.0552 ** (0.0266) | −0.0089 (0.0224) | −0.0586 ** (0.0266) |
| lnavgwag_emp | −0.0024 (0.0080) | −0.0066 (0.0053) | 0.0121 *** (0.0044) | −0.0071 (0.0053) |
| constant | 6.5165 *** (0.0737) | 0.9264 *** (0.0826) | 0.4089 *** (0.0410) | 1.0532 *** (0.0491) |
| City | Yes | Yes | Yes | Yes |
| Time | Yes | Yes | Yes | Yes |
| $R^2$ | 0.7279 | 0.4859 | 0.7253 | 0.4858 |
| *n* | 4416 | 4416 | 4416 | 4416 |

Note: Values in parentheses are standard deviations; *, **, and *** indicate significance at the 10%, 5%, and 1% significance levels, respectively.

*4.5. Heterogeneity Analysis*

China has a vast territory and abundant resources. Due to the influences of geography, history, and regional location, different cities might have very different resource elements, natural environments, and market environments. Therefore, is the policy effect of eco-industrial parks on the green development of cities different in different cities? For this reason, the heterogeneity of the policy effect of eco-industrial parks was investigated from the perspective of urban locations and urban characteristics in this study.

4.5.1. Heterogeneity of Urban Locations

Based on the heterogeneity of urban locations, with reference to Peng et al. [95], cities were first divided into cities in eastern China and cities in central and western China according to their natural location. The regression results are shown in Model (1) and Model (2) in Table 8. The pilot policy can promote improvements in the green development level of cities in eastern China, and the results are significant at the 1% level. However, the pilot policy has an insignificant influence on the green development level of cities in central and western China, and the regression coefficient of the policy dummy variable (treatpost) is negative. This indicates that implementing the pilot policy might have some negative influences on cities in central and western China, which might be due to the following reasons. Since the reform and opening-up policy, the location-oriented policies started early and developed quickly in eastern China, represented by special economic zones and traditional development zones; thus, the area accumulated more experience. This provides a prerequisite for the construction of eco-industrial parks. Moreover, eastern China has a relatively solid economic basis, abundant resource endowment, relatively complete industrial structures, and advantageous innovation conditions, which are conducive to energy restructuring and technological innovation. All of these provide the basic guarantees allowing for eco-industrial parks to the promote green development of cities. However, the economic foundation of the central and western regions is relatively weak, and it is often necessary to promote economic growth using relatively backward industries with high energy consumption, furthering social development, and laying an economic foundation

for green development. However, if certain economic conditions are not met, as well as particular social infrastructure conditions and governance levels, and the construction of large-scale ecological industrial parks is rushed, On the one hand, strict environmental regulations will inhibit the enthusiasm of some energy-consuming industries to relocate to the west, which will adversely affect the economic development of the central and western regions, with a poor business environment and insufficient resource endowment. The weak economic environment, factor environment, and governance environment make it difficult to give full play to the green policy dividends of eco-industrial parks, resulting in a loss of factor resource allocation efficiency and failure to achieve green urban development.

Secondly, city samples were divided into cities in southern China and cities in northern China according to heating supply. Specifically, cities with a heating supply are cities in northern China, while cities without a heating supply are cities in southern China. Regression results are shown in Model (3) and Model (4) in Table 8. According to the results, eco-industrial parks can promote the green development of cities in both Southern and Northern China at the 1% level, but the policy effect of cities in northern China is slightly stronger than that in cities in southern China. This might be because cities in northern China mainly use thermal power generation. In particular, the combustion of fossil fuels for heating in winter may produce pollution gases. Moreover, industries in cities in northern China are mainly "heavy" industries with serious pollution, resulting in there being greater spaces for environmental improvement. However, industries in cities in Southern China are mainly new industries with "light" structures and thus have limited space for environmental improvement. Therefore, cities in northern China are more sensitive than those in southern China to environmental regulation policies targeting environment protection and green development.

**Table 8.** Heterogeneity test results of urban locations.

| Name of Variables | Gtfp | | | |
| --- | --- | --- | --- | --- |
| | Model (1) | Model (2) | Model (3) | Model (4) |
| | Cites in Eastern China | Cites in Central and Western China | Cites in Northern China | Cites in Southern China |
| treatpost | 0.0367 *** | −0.0192 | 0.0324 *** | 0.0267 *** |
| | (0.0066) | (0.0152) | (0.0088) | (0.0077) |
| pgdp | −0.0029 * | −0.0022 | −0.0033 * | −0.0025 * |
| | (0.0015) | (0.0016) | (0.0018) | (0.0013) |
| lnsecind_gdp | −0.0059 | 0.0044 | 0.0039 | −0.0009 |
| | (0.0082) | (0.0048) | (0.0057) | (0.0060) |
| scie | −0.0087 | −0.0315 *** | −0.0727 *** | −0.0106 |
| | (0.0129) | (0.0104) | (0.0181) | (0.0098) |
| fix_pro | 0.0008 | 0.0016 | 0.0017 | 0.0018 |
| | (0.0018) | (0.0010) | (0.0013) | (0.0013) |
| FIN | −0.0010 | 0.0199 *** | 0.0125 ** | 0.0071 |
| | (0.0077) | (0.0060) | (0.0061) | (0.0080) |
| gov | −0.0457 | −0.0683 ** | −0.0828 ** | −0.0379 |
| | (0.0687) | (0.0301) | (0.0384) | (0.0378) |
| lnavgwag_emp | −0.0042 | −0.0098 | −0.0020 | −0.0131 * |
| | (0.0087) | (0.0068) | (0.0075) | (0.0075) |
| constant | 1.0829 *** | 1.0781 *** | 1.0195 *** | 1.1366 *** |
| | (0.0890) | (0.0600) | (0.0673) | (0.0709) |
| City | Yes | Yes | Yes | Yes |
| Time | Yes | Yes | Yes | Yes |
| $R^2$ | 0.4764 | 0.5029 | 0.4967 | 0.4860 |
| *n* | 1856 | 2560 | 1840 | 2576 |

Note: Values in parentheses are standard deviations; *, **, and *** indicate significance at the 10%, 5%, and 1% significance levels, respectively.



4.5.2. Heterogeneity of Urban Characteristics

Based on the heterogeneity of urban characteristics, city samples were redefined. Cities with a higher population of college students than the mean were defined as human capital cities, while cities with a lower population of college students than the mean were defined as non-human capital cities. The detailed regression results are shown in Model (1) and Model (2) in Table 9. The results demonstrated that eco-industrial parks can significantly promote the green development of human capital cities at the 1% level, and significantly promote the green development of non-human capital cities at the 10% level. The policy effect of human capital cities is more significant compared to that of non-human capital cities. This is because human capital cities possess more abundant talent reserves. As an important support for technological innovation, talents not only can transform their creative intellectual properties into new technological inventions but can also learn and absorb foreign advanced production technologies effectively, further driving technological reform. This is conducive to accelerating industrial transformation and improving the levels and application scale of clean production technologies, providing important support for the development of the policy effect of eco-industrial parks.

Secondly, city samples were divided according to the loan balance proportion of financial institutions at the end of the year in GDP. Cities with a higher proportion than the mean were defined as cities with a high financial level; otherwise, cities were defined as cities with a low financial level. On this basis, the policy effect of cities with different financial levels was investigated. The results are shown in Model (3) and Model (4) in Table 9. The estimation coefficient of the policy dummy variable (treatpost) in Model (3) is significantly positive at the 1% level, indicating that the pilot policy can improve the green development of cities with a high financial level. The estimation coefficient of the policy dummy variable (treatpost) in Model (4) is positive but is not significant, indicating that the construction of eco-industrial parks is beneficial to improving the green development of cities with a low financial level; however, this effect has no statistical significance. This might be because cities with a high financial level have advantages in terms of the number of financial institutions, deposit investment conversion rate of financial institutions, and rate of return on investment in the financial market. They optimize resource allocation and gain high-quality assets by taking advantage of the developed financial market, improving capital flow and utilization, and providing a sufficient financial basis and financial support in the implementation of the eco-industrial park pilot policy. Relatively speaking, regions with relatively backward financial development cannot support the implementation of the pilot policy due to the limitations of the financial scale, capital accumulation, and financial problems.

**Table 9.** Heterogeneity test results of urban characteristics.

| Name of Variables | Gtfp | | | |
| --- | --- | --- | --- | --- |
| | Model (1) Human Capital Cities | Model (2) Non-Human Capital Cities | Model (3) Cities with High Financial Levels | Model (4) Cities with Low Financial Levels |
| treatpost | 0.0275 *** | 0.0166 * | 0.0260 *** | 0.0033 |
| | (0.0081) | (0.0086) | (0.0074) | (0.0111) |
| pgdp | −0.0034 ** | −0.0049 *** | −0.0033 ** | −0.0024 |
| | (0.0017) | (0.0015) | (0.0014) | (0.0018) |
| lnsecind_gdp | −0.0077 | 0.0060 | 0.0022 | −0.0018 |
| | (0.0085) | (0.0044) | (0.0056) | (0.0056) |
| scie | −0.0168 | −0.0184 * | −0.0344 *** | 0.0120 |
| | (0.0133) | (0.0104) | (0.0105) | (0.0149) |
| fix_pro | −0.0051 ** | 0.0026 *** | 0.0006 | 0.0010 |
| | (0.0025) | (0.0009) | (0.0012) | (0.0013) |

**Table 9.** *Cont.*

| Name of Variables | Gtfp | | | |
|---|---|---|---|---|
| | Model (1) Human Capital Cities | Model (2) Non-Human Capital Cities | Model (3) Cities with High Financial Levels | Model (4) Cities with Low Financial Levels |
| FIN | −0.0197 ** | 0.0218 *** | 0.0031 | 0.0071 |
| | (0.0091) | (0.0055) | (0.0055) | (0.0080) |
| gov | −0.0191 | −0.0780 *** | −0.0670 ** | −0.0379 |
| | (0.0690) | (0.0293) | (0.0321) | (0.0378) |
| lnavgwag_emp | 0.0096 | −0.0110 * | 0.0064 | −0.0131 * |
| | (0.0109) | (0.0058) | (0.0076) | (0.0075) |
| constant | 0.9823 *** | 1.0876 *** | 0.9472 *** | 1.1366 *** |
| | (0.1029) | (0.0530) | (0.0707) | (0.0709) |
| City | Yes | Yes | Yes | Yes |
| Time | Yes | Yes | Yes | Yes |
| $R^2$ | 0.4473 | 0.4996 | 0.4967 | 0.4860 |
| $n$ | 1251 | 3165 | 1840 | 2576 |

Note: Values in parentheses are standard deviations; *, **, and *** indicate significance at the 10%, 5%, and 1% significance levels, respectively.

## 5. Conclusions and Enlightenments

Faced with the dilemma of both economic development and ecological environmental protection, the construction of ecological industrial parks is expected to become the key to relieving this situation. This paper takes the construction of eco-industrial parks as a quasi-natural experiment. Based on the panel data of 276 prefecture-level cities in China from 2004 to 2019, the multi-stage difference–difference method was adopted to alleviate the possible endogenous problems, and the causal relationship between eco-industrial parks and urban green development was evaluated and identified. The following three conclusions can be drawn: Firstly, under the assumption of a parallel trend, the construction of eco-industrial parks has a significant promoting effect on the improvement in urban green development level, and this conclusion is still valid under a series of robustness tests, such as placebo and PSM-DID, excluding other relevant forms of policy interference. Secondly, eco-industrial parks can help the green development of cities through three channels: improving the level of green innovation, promoting the optimization of the industrial structure, and strengthening environmental regulation. Thirdly, the green development effect of eco-industrial parks is more significant in eastern cities, northern cities, human capital cities, and high-financial-level cities, while central and western cities, southern cities, non-human capital cities, and low-financial-level cities enjoy limited policy dividends. Based on the above research conclusions, some conclusions can be drawn.

Firstly, the construction of eco-industrial parks should continue to be promoted and policy support should be strengthened. The construction of eco-industrial parks cannot only protect the regional ecological environment and realize green growth but can also promote local industrial structural upgrading and realize the dual goals of ecological environmental protection and sustainable economic development. In view of practical situations, a few cities have established eco-industrial parks in China and the construction of eco-industrial parks has not received proper attention. Hence, China should further increase publicity regarding the construction of eco-industrial parks in future policy design and encourage other qualified industrial parks to take the initiative to apply for and transform into eco-industrial parks. Moreover, the support of policies related to eco-industrial parks, such as fiscal taxation, industry and commerce policies, should be increased, and infrastructure construction should be supported, to help more cities to enjoy such policy benefits.

Second, the mechanism analysis results show that the policy effect of eco-industrial parks can realize the green development of cities by improving the level of regional green innovation, optimizing the regional industrial structure, and strengthening the local environmental regulations. This requires the government to consider the dividend release

of these three aspects in the future green policy formulation process. For example, the government should speed up the construction of laws and regulations aiming to protect green innovation patents while releasing green innovation elements on the market, reducing the cost of green innovation. The government can also give enterprises corresponding tax incentives and subsidies to carry out green innovation activities. In addition, the government should also strengthen the supervision and policy implementation of ecological industrial park emission standards to avoid green development effects being lost due to the inadequate supervision of ecological industrial parks.

Third, ecological industrial parks should be constructed according to local conditions, seeking the balanced development of the region. In view of the heterogeneity of the policy effects of eco-industrial parks in cities with different locations and characteristics, local governments should formulate corresponding policy directions based on local resource endowments and development status. For example, central and western cities can properly reduce the standards needed for enterprises to enter the ecological industrial demonstration parks. The pollution requirements and purification conditions could be lowered to avoid the adverse impact of the "one-size-fits-all" strong regulatory environment on local economic development, which would require the local government of central and western cities to consider environmental protection when building an ecological industrial park. There is also a greater need to safeguard economic growth. Therefore, compared with the eastern region, urban governments in the central and western regions need to find a balance between these two points in practice. For cities with non-human capital and cities with a low financial level, the local government of this type of city needs to support these policies when building ecological industrial parks. For example, cities with non-human capital should speed up the cultivation and introduction of talent, and implement policy guarantees aiming to guarantee the treatment and development of talents. Cities with a low financial level should speed up the introduction of capital, improve the financial market and improve the market environment. In order to realize a good combination of policies and give play to the synergistic effect of policies, the policy dividends of ecological parks should be maximized.

Differing from the existing research on the relationship between eco-industrial parks and green development, this paper focuses on the city level. On the basis of constructing multi-dimensional indicators and comprehensively measuring the level of urban green development, it tries to combine theory with actual empirical data. The relationship between the two, and the multiple mechanisms of action, are used to broaden the research ideas, whereas existing research mostly uses a single indicator to measure green development. The empirical results show that the construction of eco-industrial parks significantly promoted the green development of cities, providing a decision-making basis for countries around the world to achieve the Sustainable Development Goals. In addition, as there is a certain spatial spillover effect in the construction of eco-industrial parks, policy influence spills over to surrounding cities through learning effects and demonstration effects and further promotes the green development of neighboring cities. This is a point that has not been further discussed in this study, and there are certain limitations. Whether the implementation intensity of the pilot policy has strengthened the green development effect of eco-industrial parks is also an interesting topic, and this will be the direction of future research.

**Author Contributions:** Conceptualization, X.H.; methodology, B.L.; software, B.L.; validation, X.H., and B.L.; formal analysis, X.H.; investigation, X.H. and B.L.; resources, X.H.; data curation, X.H.; writing—original draft preparation, B.L.; writing—review andediting, X.H.; visualization, X.H. and B.L.; supervision, X.H.; project administration, X.H.; fundingacquisition, X.H. All authors have read and agreed to the published version of the manuscript.

**Funding:** This research was funded by National Social Science Fund Youth Program of China, grant number 19CJY055, and Anhui University of Finance and Economics Postgraduate Scientific Research Innovation Fund Project, funding number ACYC2022442.

**Institutional Review Board Statement:** Not applicable.

**Informed Consent Statement:** Not applicable.

**Data Availability Statement:** Publicly available datasets were analyzed in this study.

**Acknowledgments:** The authors would like to thank the anonymous reviewers for their valuable comments.

**Conflicts of Interest:** The authors declare no conflict of interest.

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
