# Peer review of "A Study on the Influence of Green Industrial Policy on Urban Green Development: Based on the Empirical Data of Ecological Industrial Park Pilot Construction"

_sustainability, doi:10.3390/su151310065_

Round 1

Reviewer 1 Report

This study uses a multi-period difference-in-differences (DID) model to explore the causal relationship between the eco-industrial park policy and urban green development. Overall, the article has a complete structural setup, but there are a series of issues to be addressed in terms of content.

1. The abstract of the article is too lengthy. The abstract is not an exact repetition of the conclusions, but a condensation of the core ideas and novelty studies of the article. In addition, it is suggested to add specific values to more clearly illustrate the impact of the eco-industrial park policy on urban green innovation.

2. The specification of Figure 2 is not enough, and there are non-English expressions in the figure legend.

3. There are more detail issues in the article. For example, the measurement criteria of control variables mentioned in 3.2.4 should be reflected in Table 3, the descriptive statistics in 3.4 are represented in Table 4, etc. The tables and textual expressions do not correspond to each other, and author should check the whole text carefully.

4. The results of the benchmark regression lack discussion. Author should compare the results of the benchmark regression with the findings of previous studies to explore the similarities and differences that exist, thus highlighting the innovation of this study. In addition, the article should also discuss the sign and significance of the control variables under the broader literature, such as why the increase in scientific and technological level inhibits urban green development, this should be explained.

5. Regional heterogeneity is not discussed enough for central and western cities. The authors do not explain clearly why the eco-industrial park policy has no effect on the green development of the central China and western China. Although the development of central and western cities is weaker than that of eastern regions, so why the result is not significant?

In general, the discussion section of the article is weak and there are some problems with the details that the author should revise carefully. In addition, the writing level of this article needs to be further improved.

Author Response

Please check the attached response.

Reviewer 2 Report

Review report on Study on the influence of green industrial policy on urban 2 green development: Based on the empirical data of ecological 3 industrial park pilot construction

I have gone through the manuscript and found it fit for the scope of this journal. The issues investigated are very topical, relevant and stand the potential of advancing knowledge in the extant literature. This is especially considering the evolving global attention on the roles of green policy in the rising devastating effects of global warming on the present ecosystem occasioned by various factors. I provide the following comments for the authors to consider in further improving the study.

1.     The abstract is well written. Kudos to the authors on this. However, I will suggest at least one novelty of the study should be stated. Similarly, the methods of analysis should be stated for the benefits of the readers. 

2.     The introduction is well written however, certain basics are lacking in it. I provided a few as follows; (i) some factual statements are not referenced. For instance, the opening paragraph has only one reference which is largely insufficient. Similar error is noted in the second paragraph (ii) the objective of this study is vague and ambiguous. It should be made explicit and concise. More so, one may find himself lucky to locate the research question which are beautifully stated by the authors. I will suggest a separate paragraph for both research questions and objectives.  (iii) Contributions to knowledge is lacking. Why? Authors should provide the contributions to knowledge. For instance, how does this study relate with certain SDGs? How will it improve the current environmental tragedy in the BRICS economies? (iii) Research about environmental sustainability in the current period should provide feedbacks from the recent development in green policies such green finance, green energy, and green technological innovation. The following studies will assist:  https://doi.org/10.1177/0958305X231177736;  https://doi.org/10.1080/13504509.2022.2162147;  https://doi.org/10.1177/0958305X2311539

3.     The literature review was merged with the introduction. How much as I will prefer a separation for rigorous review, I think the present review is far from being acceptable as there are only a few studies were reviewed.

4.      Please, create a paragraph at the end of the review to summarily appraise the reviewed studies and position the lacunas your study stands to fill.

5.     The empirical model is well stated. However, I recommend that authors state which study they follow in specifying the empirical model for the study.

6.     The methodology section fails to provide the theoretical foundation and hypotheses guiding the nexuses among the indicators. Authors should a section to discuss this key section.

7.     The discussion of the results should be linked to recent studies.

8.     Conclusion can be improved.

9.     The policy recommendations are weak in their present form. They can strengthen it by relating the findings to the recommends.

10.  Limitation and future research opportunity are missing. They should be considered.

Overall, the paper is full of potentials that can be evident if the comments above are addressed.

None

Author Response

Please check the attached response.

Reviewer 3 Report

* Organization of whole paper from abstract to conclusion need to be improved.
* There are several typo mistakes and grammatical issues throughout the manuscript.
* Methods used in this manuscript are not clearly implemented.
* Results and discussion portion needs more comprehensive explanations based on study point.
* Tables should be revised in an articulated way.

Consider these papers in your manuscript:

Maturity evaluation of supply chain procedures by combining SCOR and PST models

Using fuzzy DEMATEL and fuzzy Similarity to develop a human capital evaluation model

Mathematical modeling of Green closed loop supply chain network with consideration of supply risk: Case Study

* Organization of whole paper from abstract to conclusion need to be improved.
* There are several typo mistakes and grammatical issues throughout the manuscript.
* Methods used in this manuscript are not clearly implemented.
* Results and discussion portion needs more comprehensive explanations based on study point.
* Tables should be revised in an articulated way.

Consider these papers in your manuscript:

Maturity evaluation of supply chain procedures by combining SCOR and PST models

Using fuzzy DEMATEL and fuzzy Similarity to develop a human capital evaluation model

Mathematical modeling of Green closed loop supply chain network with consideration of supply risk: Case Study

Author Response

Please check the attached response.

Reviewer 4 Report

Dear Authors

After detailed readings in the manuscript, entitled: "Study on the influence of green industrial policy on urban green development: Based on the empirical data of ecological industrial park pilot construction". The manuscript is scientifically fascinating as it presents a vision of economic development and ecological environmental protection, focused on the construction of eco-industrial parks, based on the panel of 276 prefectural-level cities in China from 2004 to 2019, the causality between eco-industrial parks and the green development of cities, showing a focus on sustainability. Therefore, I suggest the ACCEPTANCE of the manuscript with minor corrections:

1 - Could clarify at the end of the Abstract the importance of this study on a global scale. This would arouse greater interest from the journal readers.

2 - In the "Keywords" I suggest the insertion of the following terms: sustainability; quality of life.

3 - The following sentence in the introduction (Hence, heterogeneity of policy effect of eco-industrial parks is further discussed in this study) (lines 160 - 161) could be completed with the importance of this discussion at a global level for society.

4 - I suggest allocating higher resolution in Figure 1.

5 - I suggest allocating higher resolution in Figure 2.

6 - I really liked the approaches and discussions presented in the results. The authors did a good job. Congratulations.

7 - In conclusion this sentence is confusing: "Therefore, the article intends to discuss from this dimension in the next step. At the same time, whether the implementation of pilot policies has strengthened the green development effect of eco-industrial parks is also an interesting point of consideration" (line 721 - 723). I suggest its withdrawal.

8 - In conclusion, it is necessary to talk more about the possibility of future studies to be applied on a global scale. This would enable an idea of scientific continuation.

9 - The English of the manuscript is very well written. Congratulations to the authors.

Author Response

Please check the attached response.

Reviewer 5 Report

The objective of the paper is finding the causality between eco-industrial parks and green development of cities. The authors using a panel data of 276 prefecture-level cities in China from 2004 to 2019.

A lot of text without any reference (e.g.: line 56 to 96; line 101 to 110; line 220 to 242…)

The hypotheses could be proposed in the end of 2.1, 2.2 and 2.3 and not only the end of 2.3.

The “control variable” must be explained in the for formula (1)

E.g., The “control variable” consisting in the factors influencing green development.

Author Response

Please check the attached response.

Round 2

Reviewer 1 Report

I think the paper has potential for publication. The authors should revise the paper as it includes some typos.

Author Response

Point 1: I think the paper has potential for publication. The authors should revise the paper as it includes some typos.

Response 1: Thank you very much for the reviewer's recognition of the article. Thanks for this, the author has carefully checked and modified the spelling of the whole text according to the reviewer's suggestion. Thank you very much for your valuable advice.

Reviewer 3 Report

accept

Author Response

Point 1: accept

Response 1: Thanks for the reviewer's affirmation. Your affirmation is our greatest encouragement. I would like to express my heartfelt thanks to you.